# Recycled arc mantle recovered from the Mid-Atlantic Ridge

B. M. Urann [1✉], H. J. B. Dick [2], R. Parnell-Turner[3] & J. F. Casey[4]

Plate tectonics and mantle dynamics necessitate mantle recycling throughout Earth's history, yet direct geochemical evidence for mantle reprocessing remains elusive. Here we present evidence of recycled supra-subduction zone mantle wedge peridotite dredged from the Mid-Atlantic Ridge near 16°30′N. Peridotite trace-element characteristics are inconsistent with fractional anhydrous melting typically associated with a mid-ocean ridge setting. Instead, the samples are best explained by hydrous flux melting which changed the melting reactions such that clinopyroxene was not exhausted at high degrees of melting and was retained in the residuum. Based on along-axis ridge depth variations, this buoyant refractory arc mantle is likely compensated at depth by denser, likely garnet-rich, lithologies within the mantle column. Our results suggest that highly refractory arc mantle relicts are entrained in the upper mantle and may constitute >60% of the upper mantle by volume. These highly refractory mantle domains, which contribute little to mantle melting, are under-represented in compilations of mantle composition that rely on inverted basalt compositions alone.

[1] MIT-WHOI Joint Program, Marine Geology and Geophysics, Woods Hole Oceanographic Institution, Woods Hole, MA 02543, USA. [2] Woods Hole Oceanographic Institution, Woods Hole, MA 02543, USA. [3] Institute of Geophysics and Planetary Physics, Scripps Institution of Oceanography, La Jolla, CA 92093, USA. [4] Department of Earth Sciences, University of Houston, Houston, TX 77204, USA. ✉email: burann@whoi.edu

Mantle geochemistry involves the study of stochastic occurrences of peridotite from abyssal settings, xenoliths, and ophiolites. The random nature of sample acquisition requires certain assumptions and extrapolations to make meaningful interpretations of the mantle writ large. In classical theory, fertile mantle peridotite undergoes adiabatic melting during ascent beneath a divergent plate boundary; after melting, the restite will be depleted in FeO and incompatible elements, resulting in a refractory residue that is more buoyant than the primitive mantle from which it was derived[1]. While early studies relied on basalt compositions to infer source characteristics, and often assumed a homogeneous mantle, subsequent studies of mantle peridotites have shown unequivocally that the mantle is heterogeneous on length scales from meters to thousands of kilometers with respect to isotopes, major elements, and trace elements (e.g. Figs. 1 and 2)[2–9]. These heterogeneities reflect a time-integrated aggregate of inherited heterogeneity during Earth's accretion, dynamic advection of material, localized refertilization and re-equilibration, and mantle melting at both divergent and convergent plate margins over its history. Re–Os and Pb isotopic dating of abyssal peridotites has provided further insights into ancient (up to 2.0 Ga) melt depletion events[10,11], however, relating peridotite trace element characteristics to previous melting events remains difficult and non-unique. If mantle advection is indeed robust, relicts of previous melting events are expected to occur in dissimilar tectonic settings: to date, evidence for this concept is scarce. In one instance, Simon et al.[12] found that nearly two thirds of mantle xenoliths sampled from ocean island basalts represent ultra-refractory harzburgites (past clinopyroxene exhaustion) that resemble forearc peridotites. Gao et al.[13] used Al-depleted peridotite bulk rock compositions to suggest that residues of hydrous melting could be entrained beneath the Southwest Indian Ridge, supporting the topographic relief observed along the Marion Rise. Nd and Hf isotope systematics in basalts and olivine hosted melt inclusions have provided additional evidence for ancient, ultra-depleted mantle domains beneath the Azores Rise[14,15]. These results show that our understanding of mantle composition and how it relates to basalt chemistry at slow and ultra-slow spreading mid-ocean ridges needs major modification, particularly where ultramafic samples appear to be uncorrelated to local basalt compositions[4,9].

The 14°–17°N region of the Mid-Atlantic Ridge (MAR) has long been an enigma for mantle geochemists. This slow-spreading region (~25 km Ma$^{-1}$[16]) contains refractory, trace-element depleted peridotites (indicative of high degrees of melting) exposed directly on the seafloor, yet basalts recovered in the area display enrichments in incompatible and light rare earth elements with respect to normal mid-ocean ridge basalt (MORB) as well as high $Na_{8.0}$ values, suggesting low degrees of melting[5,17–24]. The decoupling between peridotite and MORB major and trace element chemistry is likely a consequence of differential upwelling velocity between melt and the solid source, however recent workers have attributed to this to inherited mantle heterogeneity whereby more fertile (and radiogenic) domains contribute disproportionately large melt volumes to MORB compared to refractory mantle domains[2,25–27]. The provenance of these highly refractory domains remains an open question, since melting at a mid-ocean ridge is generally limited by the thickness of the lithospheric lid and the clinopyroxene-out peritectic point[28], consequently restricting melting at slower spreading ridges[29–32]. Here we use clinopyroxene (Cpx) from 16°N abyssal peridotites to show that hydrous melting is required to explain their highly depleted heavy rare earth element (HREE) and Ti concentrations, suggesting that parcels of entrained mantle are in fact recycled from subduction zones to mid-ocean ridges and are likely a ubiquitous mantle component.

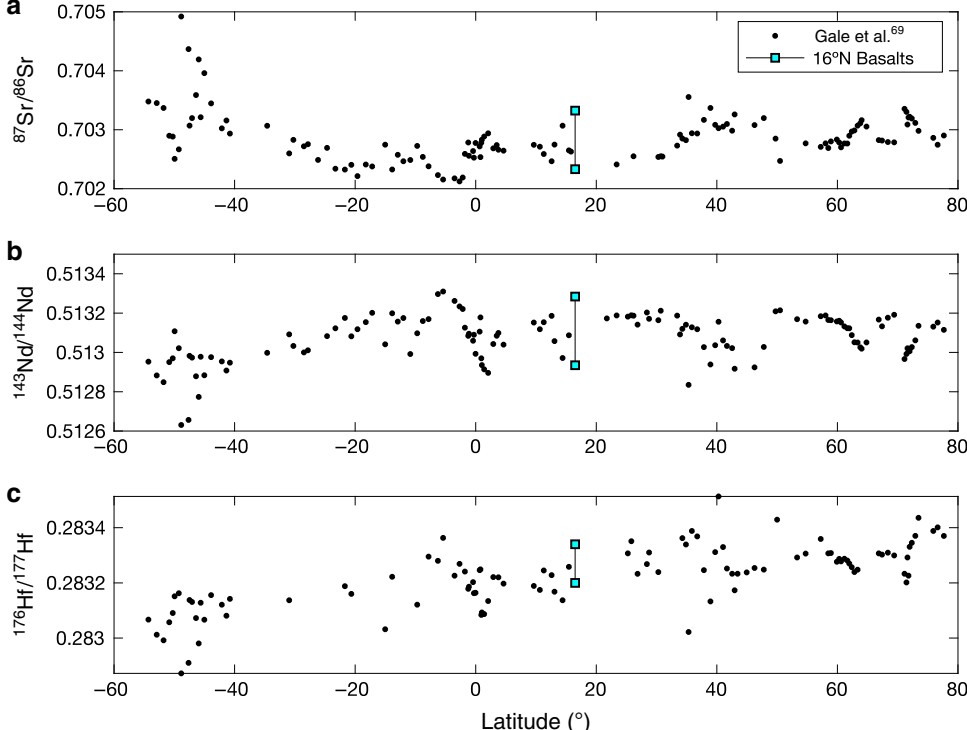

**Fig. 1 Mid-ocean ridge basalt isotopic variability along the Mid-Atlantic Ridge.** MORB segment averages of $^{87}Sr/^{86}Sr$ (**a**), $^{143}Nd/^{144}Nd$ (**b**) and $^{176}Hf/^{177}Hf$ (**c**) from Gale et al.[69] exemplify the time-integrated source heterogeneity observed in Mid-Atlantic Ridge basalts. 16°N region, labeled, shows the heterogeneous nature of source material on scales of tens of kilometers, typical for Atlantic MORB. 16°N samples are given as a range with data from Henrick[70].

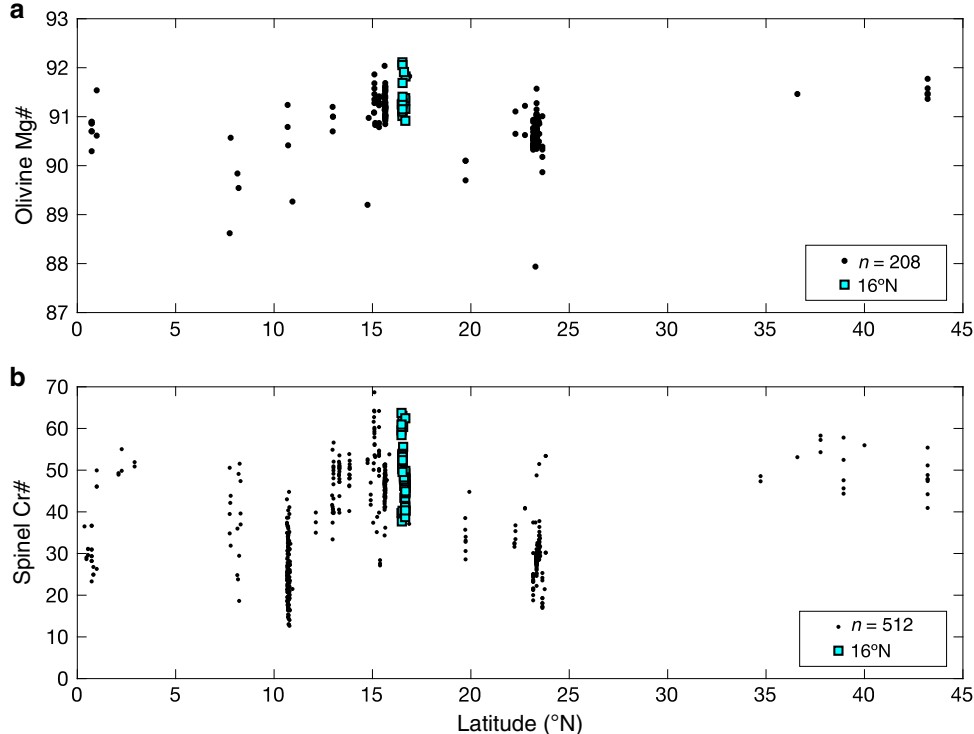

**Fig. 2 Mid-Atlantic Ridge latitudinal variations in peridotite composition. a** Olivine Mg# (molar Mg/(Mg + Fe²⁺)) along the Mid-Atlantic Ridge shows the anomalously refractory compositions present at the 14–17°N region. Additional data from 15°20′N FZ in Supplementary Data 4, previously unpublished. **b** Peridotite spinel Cr# (molar Cr/(Cr + Al)) extend to very high values (>55) between 14° and 17°N, atypical of anhydrous melting at mid-ocean ridge settings. Black data points for both plots from the literature[23,42,71]. Analytical uncertainties smaller than symbols.

## Results

**Regional synopsis and sample description.** *R/V Knorr* Cruise 210 Leg 5 dredged residual mantle harzburgite, basalt, and gabbros along a 50–km long segment of the Mid-Atlantic Ridge between 16°15′N and 16°50′N that encompasses a southern fully magmatic, a central nearly amagmatic, and a northern weakly magmatic series of core complexes. Detailed bathymetry, descriptions, and dredge data can be found in Smith et al.[33]. Twenty harzburgite samples were selected for major element analysis of spinel, pyroxenes, and olivine. Samples have olivine Mg# (100 × Mg/[Mg + Fe²⁺]) of 91 to 92.1 and chromian spinel Cr# (100 × Cr/[Cr + Al]) of 37–64 (Figs. 2 and 3, Supplementary Data 1). Point counting shows very low but discernable primary Cpx abundances, whereas some samples appear be devoid of Cpx irrespective of secondary alteration. Two particularly fresh, Cpx-bearing harzburgites were chosen for trace element analysis. Both samples (D41–24, D41–82) display protogranular textures with interstitial Cpx in textural equilibrium with olivine, orthopyroxene (Opx), and chromian spinel (Supplementary Fig. 1). Cpx appears to be primary in origin based on textural observations, and not formed as an exsolution feature. Pyroxene grain sizes range from 500 to 3000 μm and were analyzed for trace elements using laser ablation inductively coupled plasma mass spectrometry (see "Methods"). Trace element concentrations are shown in Supplementary Data 2. Additional, previously unpublished Cpx REE data from the Fifteen-Twenty Fracture Zone is provided in the supplementary materials (Supplementary Data 4). Samples are ultra-depleted in HREE, middle rare earth elements, and high field strength elements (HFSE), whereas light rare earth elements (LREE) define a shallower slope than one would expect from fractional melting models based on lanthanide compatibility (Supplementary Fig. 2). LREE decoupling from HREE and HFSE may be a result of partial re-equilibration with migrating melt in

the mantle column as it ascended[34], a product of disequilibrium melting (e.g. Liang and Liu[35], Supplementary Fig. 3), or have been generated by addition of an LREE-enriched subduction-derived fluid. Our study therefore focuses on HREE and Ti, which are fluid immobile and less susceptible to diffusive re-equilibration with passing melts during emplacement.

## Discussion

**Provenance of 16°N peridotites.** Cpx analyses presented here are more HREE and Ti depleted than any previously reported abyssal peridotites. These samples bear a striking geochemical resemblance to supra-subduction zone (SSZ) peridotites from convergent margins formed under hydrous conditions with respect to Cpx trace element concentrations (Supplementary Figs. 4 and 5) and olivine and spinel major element compositions[36–39]. In particular, 16°N peridotites display extreme Cr# spinel compositions greater than ~55 (Fig. 3), values that are normally restricted to supra-subduction zone settings (e.g. the Josephine Peridotite, Oregon, USA)[40]. 16°N harzburgites also show similar Opx Al₂O₃ and spinel Cr# to highly refractory peridotite xenoliths sampled from ocean island basalts and described by Simon et al.[11], as well as other SSZ peridotites[38,39] (Supplementary Fig. 6). To better understand the genesis of our samples, we modeled non-modal fractional melting of a depleted MORB mantle in the garnet and spinel stability field under anhydrous conditions typical of a mid-ocean ridge setting after Hellebrand et al.[41] (and references therein, see Methods) and compared our results with published abyssal peridotite data (Fig. 4)[42]. Published data show a relatively narrow range of Cpx trace element concentrations in Yb–Ti space. Anhydrous melting alone is not sufficient to achieve the concentrations observed in our 16°N samples, as Cpx is exhausted before such depleted compositions

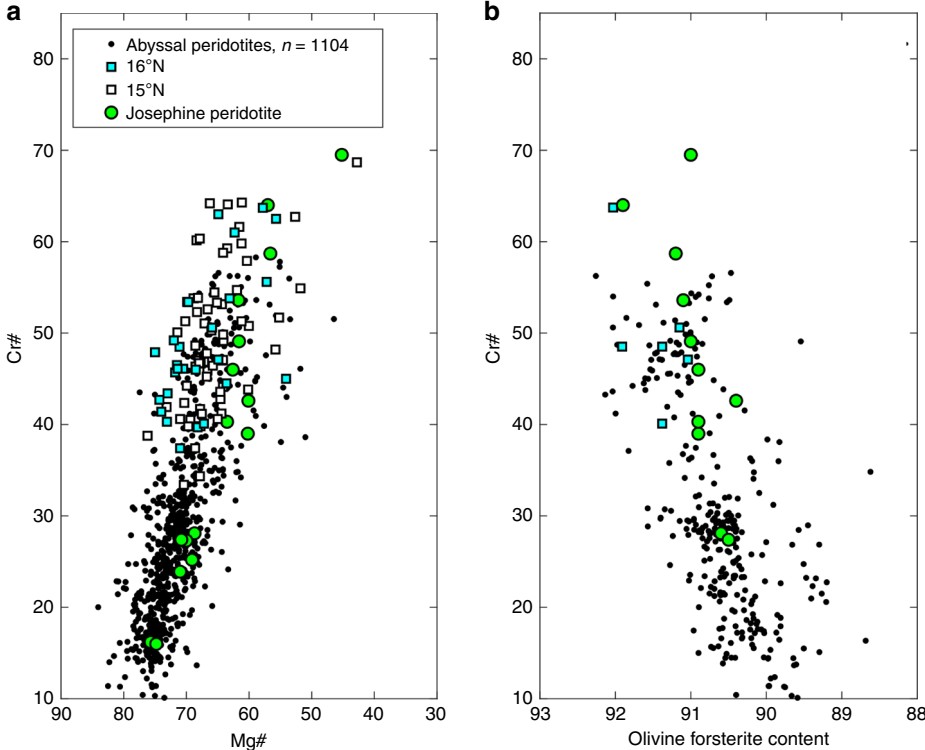

**Fig. 3 Chromian spinel and olivine compositions of 14–16°N peridotites compared to abyssal peridotites. a** Cr# after Dick et al.[40]. **b** Spinel Cr# plotted against coexisting olivine forsterite content. Josephine peridotite, a well-known SSZ locality, plotted for comparison with data from Le Roux et al.[39]. Abyssal peridotite data from the compilation of Warren[42]. Not all spinel–olivine pairs were analyzed due to alteration. Analytical uncertainties smaller than symbols.

are reached (Fig. 4, white circles and hexagons). Even if the peridotite were to undergo two distinct anhydrous events (e.g. 20% melting followed by a second anhydrous event), such concentrations would still not be attained before Cpx was exhausted from the residue by additional melting during the second event. Interestingly, EPR harzburgites from Hess Deep reported by Dick and Natland[43] also appear to reside in the SSZ field, with Cr-spinel values (Cr#) of more than 50 and highly depleted Cpx trace element abundances, warranting further investigation.

Fractional melting under hydrous conditions is known to decrease the proportion of Cpx entering the melt with respect to Opx[36,44]. Hydrous melting therefore provides an efficient means of retaining Cpx in the residuum to higher extents of melting than anhydrous conditions allow, permitting higher degrees of depletion with respect to incompatible elements (e.g. HFSE). We model such a process in Fig. 4 (blue squares), where 16°N Ti and Yb systematics are reproduced by ~8% hydrous melting in the garnet stability field, followed by an additional high degree (>20%) of hydrous melting in the spinel field. 16°N harzburgites may have melted further upon emplacement underneath the MAR, although mineral modal abundances suggest this would have been limited to retain any primary Cpx. Overall, their low HREE and HFSE incompatible element abundances, mineral modes and compositions make them the most refractory abyssal peridotites yet studied from a mid-ocean ridge setting. We also calculated melts in equilibrium with 16°N Opx based on published partition coefficients; the trace element characteristics, in particular their fluid-mobile element enriched and Ti depleted nature, are most consistent with a boninitic melt, similar to those found presently in the Bonin Archipelago (Supplementary Fig. 5)[45]. Boninites have characteristically low HFSE abundances coupled with fluid-mobile element and LREE enrichments; a boninite-like melt can reconcile both fluid-mobile element and

LREE enrichments while maintaining Ti depletions as seen in 16°N Cpx and Opx (Supplementary Fig. 5). It is notable then that our sample set includes several orthopyroxenite veins, always dredged along with dunite, but rare in other abyssal peridotite suites, yet consistent with an origin as late boninitic melts produced during the latter end of sub-arc melting as described in other SSZ mantle provinces[38,46,47].

Given the long history of subduction, accretion, and continental breakup, ultra-refractory harzburgitic parcels formed in arc-mantle wedge settings are likely entrained throughout the mantle; more detailed sampling could identify similar localities throughout the mid-ocean ridge system. To that end, we interpret the 14°–17°N MAR region as a parcel of buoyant arc mantle captured in the upwelling mantle column, similar to, but more refractory than peridotites described by Seyler et al.[48] along the Southwest Indian Ridge. This refractory mantle province is likely a relict of a Proterozoic melting event as inferred by Re–Os isotope systematics at the 15°20′N region[11]; our results expand on this, and suggest a subduction zone setting. LREE-enriched basalts with relatively high $Na_{8.0}$ dredged in the region are therefore primarily sourced from more fertile components (e.g. lherzolite or pyroxenite veins), and are not representative of melting the modal or chemical composition of the shallow sub-ridge mantle[21]. Regional basalt compositions require a more fertile, possibly denser source mantle deeper in the melting column; a necessity required to explain the great depth of the ridge (Fig. 5). The incongruence between peridotite compositions (residues) and mantle melts (MORB) suggest that variable source contributions must be carefully considered when evaluating regional MORB geochemistry.

**A heterogeneous Atlantic mantle.** Although mantle peridotite at 14–167°N is highly refractory (and therefore buoyant), the region

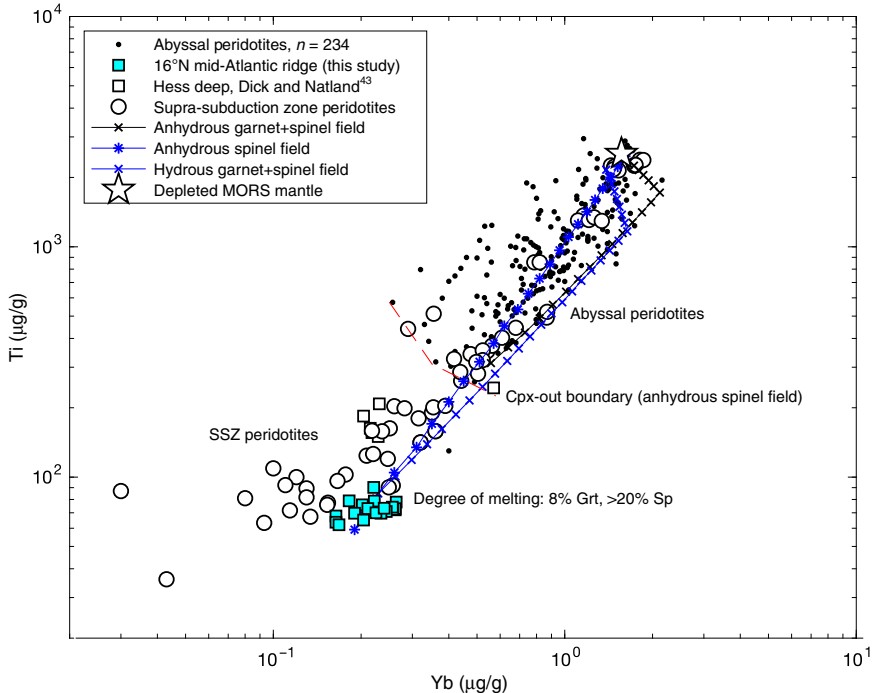

**Fig. 4 Fractional melting model of incompatible elements titanium and ytterbium in clinopyroxene.** Anhydrous melting alone cannot replicate the ultra-refractory Cpx compositions before Cpx is exhausted, whether one considers spinel field or garnet followed by spinel field (black ticked line) melting. However, changes in melting modes and partition coefficients in a hydrous melting environment permit extreme depletions in Ti, matching observations (blue lines). Cyan squares represent each individual measurement from 16°N Cpx (Supplementary Data 2). Red dashed line is approximate spinel-field Cpx-out boundary based on modeled anhydrous melting of DMM spinel bearing peridotite. Model details, including melting modes and partition coefficients, can be found in the Methods section and Supplementary Data 5. SSZ peridotites are thought to have undergone extreme degrees of melting in a mantle wedge setting, often resulting in the exhaustion of Cpx; SSZ Cpx Ti and Yb abundances are very similar to those observed in 16°N Cpx. Abyssal peridotite data from the literature as compiled by Warren[42]. SSZ peridotite data from the literature[36–39].

is not associated with a rifted bathymetric rise similar to Marion or the Azores (i.e. Zhou and Dick[9]). Rifted rises such as Marion are a function of both compositional heterogeneity and to a lesser extent thermal variations; thermal variations are thought to be modest (<50 °C) throughout the upper mantle[1,9,49,50]. The density contrast due to chemical heterogeneity between refractory (harzburgitic) and fertile (lherzolitic) mantle components is negligible in the spinel facies (<70 km), however, below ~70 km this density contrast increases markedly up to 1.2% within the garnet facies since bulk rock densities are sensitive to garnet modal proportions and hence, bulk $Al_2O_3$ content[1]. Our preferred interpretation is that the entire region, spanning ~3° of latitude (330 km), contains entrained parcels of buoyant arc mantle that is isostatically compensated at depth by more fertile, dense components (e.g. subducted ocean crust or fertile MORB mantle). At a given pressure and temperature, eclogite (subducted oceanic crust) will have a higher density and seismic shear wave velocity than peridotite[51], providing a means of assessing the feasibility of such a model. In fact, our hypothesis is consistent with tomographic observations, which show positive shear wave velocity ($V_s$) anomalies at 300 km depth beneath the ridge axis in the 14–17°N region and centered below the Vema Fracture Zone at 11°N (Fig. 5)[52,53]. The positive relationship observed between ridge depth and $V_s$ observed along the Mid-Atlantic Ridge at 300 km depth (Fig. 5) may be explained by either thermal anomalies or compositional heterogeneities (e.g. Dalton et al.[54]). However, the coefficient of thermal expansion for peridotite is modest; for a constant composition, a mantle potential temperature difference of 50 °C (1320 °C versus 1370 °C) yields less than 500 m of relief in a 300 km column based on the parameterization of Abers and Hacker[55]. Thus, the anomalous ridge depths (~1 km) are likely

the result of lithologic heterogeneities, manifested by significant density and compositional differences; the presence of buoyant arc mantle material in the 16°N region exemplifies the chemical diversity of the sub-ridge mantle. Here, density compensation at depth has effectively muted the ridge bathymetric high that would otherwise exist, as seen further north along the Azores rise, which is compensated by low-density refractory harzburgite (Fig. 5)[4].

Fertile material at depth, which is required to explain the elevated MORB $Na_{8.0}$ and LREE enrichments, provided the major contribution to melt genesis at 16°N due to lower solidus temperatures, while an overlying ultra-refractory component supplied a lesser amount. Isotopically, basalts in the region display extremely radiogenic εNd (8.6–11.3) and εHf (15.9–19.5) values, likely reflecting a small contribution from highly depleted mantle[21,56]. Previous MORB studies have inferred a highly depleted source to reconcile decoupling of Nd and Hf isotopic systematics and extreme Hf values, e.g. the Residual Lithosphere (ReLish) component of Salters et al.[56] or subduction-modified mantle of Janney et al.[57]. Our samples provide a likely candidate for such a ubiquitous mantle component, corroborating these studies. While isotopic analyses are beyond the scope of this work, we speculate that 16°N Cpx should possess highly radiogenic Hf isotopic compositions based on elevated Lu/Hf ratios (>1) of 16°N Cpx with respect to DMM (~0.37[58]) coupled with ancient Re–Os ages of nearby peridotites[11].

**Implications**. Mantle heterogeneity likely plays an important role in the styles of oceanic crustal accretion, dictating the amount of melt available to generate new basaltic crust if ultra-refractory parcels are present in variable proportions in the melting column.

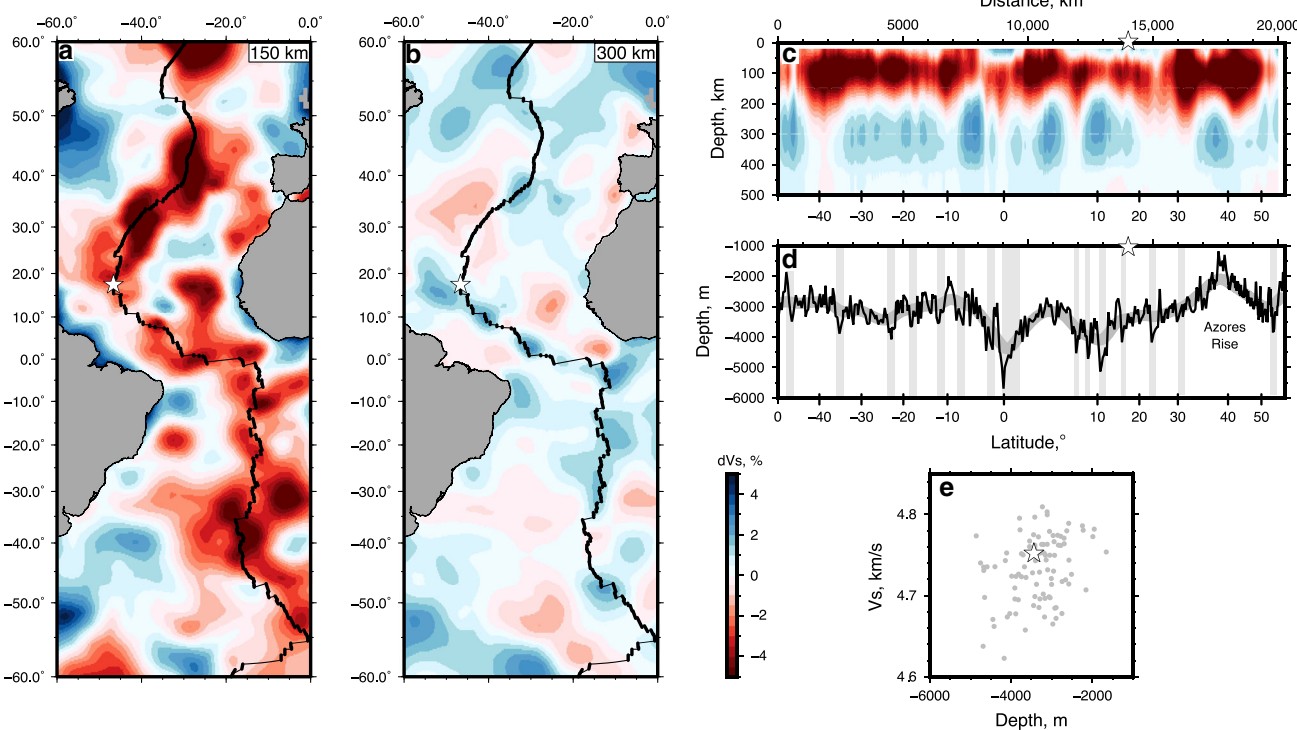

**Fig. 5 Shear wave velocity structure beneath the Mid-Atlantic Ridge. a**, **b** Horizontal slices through SL2013sv shear velocity tomographic model where percentage anomaly ($dV_s$) is with respect to AK135 model at 150 and 300 km, respectively[53]; thick/thin black lines are spreading segments/fracture zones, respectively[69]. Stars denote 16°N segment. **c** Vertical slice along MAR shown in **a**, **b**; gray bands are major fracture zones, star is 16°N segment. **d** Black/gray line is ridge depth smoothed with 2000 km Gaussian filter. **e** Ridge depth smoothed with a 280 km Gaussian filter, compared with $V_s$ (gray dots) along the MAR between 50°S and 55°N; star is 16°N segment.

**Table 1 Mantle processing calculations.**

| Earth Radius (km) | Mantle subduction zone influx (km³ year⁻¹) | Earth volume below 30 km (km³) | Mantle volume below 660 km (km³) | Mantle volume 660 to 30 km depth (km³) | Mantle processed since 2.5 Ga (km³) | Percent mantle processed since 2.5 Ga |
|---|---|---|---|---|---|---|
| 6378 | 77 | 1.07E + 12 | 7.83E + 11 | 2.88E + 11 | 1.93E + 11 | 67% |

Mantle processing calculations based on mantle subarc melting region influx of 77 km³ per year of Jagoutz et al.[60] and a volume of mantle between 30 and 660 km depth.

However, the amount of ultra-refractory mantle residing in the mantle remains unknown. Based on estimates of the volume of mantle entering the sub-arc melting region per year (77 km³ year⁻¹, Jagoutz and Schmidt[59]), and assuming this flux has been approximately constant over time, we estimate that ~67% of the mantle above the 660 km mantle discontinuity could have been processed through subduction zone environments, and thus could be considered ultra-refractory, over a period of ~2.5 Ga (Table 1). This is consistent with stochastic sampling of ocean island basalt-hosted xenoliths presented by Simon et al.[12] where two thirds of samples were considered highly refractory (Supplementary Fig. 6). We caution that our estimates represent upper bounds, as the influx of mantle entering the sub-arc region is not entirely melted under hydrous conditions. Cumulate rocks derived from both hydrous and nominally dry differentiation trends are commonly found in arc terranes, indicating both hydrous flux melting and adiabatic decompression melting (e.g. Jagoutz et al.[60]), the former of which would be responsible for generation of SSZ peridotite. However, our calculations square well with those of Salters et al.[56] who estimated up to 50% ReLish in the MORB source to reconcile basalt Hf isotopic compositions. While recycled SSZ peridotite collected from mid-ocean ridges are limited to a handful of locations at present

(14–17°N MAR and SWIR 53°E), we note that vast swaths of mid-ocean ridge system, including almost the entire southern Atlantic, have yet to be dredged off axis or studied in detail (see ref. [42] Fig. 1 for map of current sample locations).

On a more local scale, a large proportion of refractory components will induce less adiabatic melting and generate lesser volumes of melt, leading to limited magmatically-accommodated spreading and inducing tectonic extension in the form of peridotite emplacement directly on the seafloor (e.g. significant portions of the Southwest Indian, Gakkel and Mid-Atlantic Ridges[30,61,62]). This is the case at the 14–17°N region and represents an end-member mantle source composition dominated by ultra-refractory peridotite upwelling at the ridge axis.

The abundance of this refractory material will have implications for rheological and geophysical properties, as this material may have higher viscosities than more fertile mantle due to low water abundances, as well as higher shear wave velocities due to higher olivine forsterite contents[63,64]. Further study of mantle heterogeneity will require detailed sampling of mid-ocean ridges, while systematic peridotite geochemical characterization could lead to new insights as to the distribution of variably depleted source lithologies in the mantle. Buoyant, highly refractory melt

residues are likely ubiquitous in the upper mantle, congruent with the hypotheses of previous workers[51]; the former could comprise as much as 67% of the upper mantle, providing compelling evidence for ultra-depleted mantle (ReLish, subduction-modified mantle) throughout the MORB source. Mantle heterogeneity exerts a fundamental influence on melt generation, MORB composition, and plate tectonics, and must be considered insofar as mantle geochemistry and dynamics are concerned. 16°N peridotites add to a growing body of evidence, which suggests that the upper mantle as a whole is more depleted than previously thought.

## Methods

**Analytical procedure.** In all, 150 µm thick sections were analyzed for trace elements at the University of Houston utilizing the Varian 810 ICP-MS coupled with a Cetac LSX–213 Laser Ablation Microprobe. Laser data were acquired as three-spot analyses by drilling a 100 µm diameter in single-spot mode of operation, at a 20 Hz pulse repetition rate. The spots of interest were measured in mass sequence by peak hopping mode (1 point per isotope). The analytical sequence protocol includes the analysis of the gas blank for 30 s, (measured for the carrier gas without ablation at the beginning of analysis of each sample), followed by the USGS glass standard treated as an unknown and then an external calibration standard in sequence. This is successively repeated after every sample with five single-spot sample ablation measurements for each sample analyzed. Data processing of these results was conducted using the data reduction software system GLITTER 4.0 developed by Van Achterbergh et al.[65] and discussed by Griffin et al., (http://www.es.mq.edu.au/GEMOC/). The real-time data processing included correction for the gas blank and the use Mg isotopes as internal standards. Counts were normalized to MgO major element data using Glitter. The method detection limits are automatically calculated by the Glitter software as three times the standard deviation (1σ) obtained from analysis of the level of gas blank/background signal normalized to instrument sensitivity, which is the signal intensity for a given concentration, expressed as counts per second (cps), per concentration unit (ug/g). The USGS BIR-1G standard was used as the external calibration standard reference glass and USGS standard BHVO-2G as unknown glass standard. The preferred standard values are from GeoReM database (http://georem.mpch-mainz.gwdg.de/). Reproducibility (RSD) was better than five percent for all elements measured with respect to BIH-1 (Supplementary Data 3).

Major elements for individual phases (spinel, olivine, pyroxenes) were analyzed using a JEOL JXA-8200 electron microprobe at the Massachusetts Institute of Technology. A 10-nA beam current and 15 kV accelerating potential were used for all analyses. Beam diameter was 10 µm for pyroxenes along ten-spot core to rim traverses and 1 µm core to rim traverses for spinel and olivine; mean values are reported. Standards used were synthetic aluminous orthopyroxene (ALP7), synthetic diopside-jadeite (DJ35), Stillwater chromite (52NL11), and natural Marjalotti olivine (MARJ). Standard deviations were <2% for major elements. Data reduction was done using CITZAF software[66]. The counting times used for the phase analyses range was 40 s on peak, and 20 s on background.

**Trace element modeling.** We applied the non-modal fractional melting equations of Shaw[67] to both spinel and garnet peridotites under hydrous and anhydrous conditions, using the Depleted MORB Mantle (DMM) starting chemical composition of Workman and Hart[58] (Supplementary Data 5). Anhydrous garnet and spinel stability field melting models follow Hellebrand et al.[41] and references therein. While garnet field melting was included in the modeling, it is not required; 23% hydrous melting of DMM in the spinel field is able to reproduce trace element abundances, whereas anhydrous melting exhausts Cpx at 18–19% melting, depending on melting modes used. To test the robustness of our conclusions, we used a variety of melting reactions from the literature (Supplementary Data 5); anhydrous melting reactions are unable to reproduce observed trace element abundances before Cpx exhaustion. Model parameters and solutions are available in Supplementary Data 5. Hydrous melt modes in the garnet stability field from high-T (1075-1125 °C) experiments of Grove and Till[68], spinel field melting modes from Bizimis et al.[36] after Gaetani and Grove[44].

## Data availability

The authors declare that all data supporting the findings of this study are available (or cited) within the paper and supplementary data files (Supplementary Data Files 1–5). Mantle density calculations provided by the authors upon request.

## Code availability

Figures were generated using MATLAB. Code is available upon request.

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

## Acknowledgements
We thank the science party for their dutiful collection and description of dredge samples, and in particular chief scientist Dr. Deborah K. Smith. Analysis work for this research was supported by an internal grant from the MIT EAPS Student Research Fund to BMU. Urann was supported by the Stanley W. Watson Student Fellowship Fund based at WHOI. Dick and Urann were supported by NSF OCE-1637130 and OCE-1155650. Dr. Yongjun Gao is thanked for conducting LA-ICP-MS trace elements analyses.

## Author contributions
H.J.B.D. and B.M.U. initiated the project as part of B.M.U.'s PhD. B.M.U. gathered the major element data, conducted the REE modeling and wrote the initial draft of the paper. H.J.B.D. secured funding and contributed to the manuscript. J.F.C. conducted LA-ICP-MS trace element analyses and contributed to the paper. Parnell-Turner created Fig. 5 and contributed to the paper.

## Competing interests
The authors declare no competing interests.
