## [Peer Review File · Nature Communications]

Reviewers' comments:

Reviewer #1 (Remarks to the Author):

Crust and mantle recycling are a natural consequence of plate tectonics and mantle convection, and an important topic in geochemistry, geodynamics, and seismology. Typical geochemical studies of recycling rely on isotope ratios and trace element abundances in basalts from ocean islands and mid-ocean ridges, and almost all these studies focus on the recycling of crust materials as they have larger isotopic (e.g., $^{87}/^{86}\text{Sr}$) and highly incompatible trace element (e.g., LILE) signals than mantle samples. Geochemical studies of the recycling mantle materials rely largely on Nd, Hf and Os isotope ratios and the main conclusion is that the recycled mantle materials are old and refractory. There is no study that I am aware of that can pinpoint/deduce the tectonic setting of recycled mantle materials. The present study did just that by arguing that the ultra-depleted, clinopyroxene-bearing harzburgites at Mid Atlantic Ridge near 16°N representing a piece of recycled arc mantle. The authors' conclusion is based largely on similarities in major elements (Cr# in spinel, Mg# in olivine, Al_2O_3 in clinopyroxene) and selected incompatible trace elements (HREE and Ti) between their samples and peridotites from supra-subduction zone settings. They argue via trace element modeling that hydrous melting is required to explain the presence of residual clinopyroxene in the harzburgite and the very depleted Ti and HREE abundances in clinopyroxene (their Fig. 4). I generally agree with this argument and believe it is a novel study that will likely generate considerable interests in mantle geochemistry and geophysics communities. Nonetheless, I believe the modeling part can be improved (my main points #1 & #2 below) and additional data may help to strengthen or test the conclusion presented in this study (#2). Overall, the manuscript is well organized and easy to follow. I have a few suggestions on wording/structure that are listed in "Additional comments..." below. I would recommend this interesting manuscript for publication in NC after a round of revision.

Main Points:

1. Modeling of hydrous melting: The key figure used in study is Fig. 4 which compares Ti and Yb abundances in abyssal peridotites and SSZ peridotites with those predicted by anhydrous and hydrous melting. The modeling follows almost exactly the same method as Bizimis et al (2000), which is reasonable first step. But this is an old study. For example, it is not clear to me where Bizimis and coworkers got their hydrous melting reaction which you used in the present study. This is crucial in explaining clinopyroxene modal abundance and Ti-Yb depletion. Considerable progresses have been made in understanding hydrous melting since 2000 and such information should be included in the hydrous melting model in the present study. For example, the solidus of hydrous peridotite is considerably lower than the anhydrous one that hydrous melting will most likely start in the garnet stability field. There are new melting reactions for hydrous melting, the most recent example is Grove and Till (CMP 2019). You could test a number of them to demonstrate the robustness of your conclusion.
2. Mantle source composition: just curious, if you melt a garnet pyroxenite or a pyroxenite-peridotite mixed source, would it be possible to produce the ultra-depleted cpx observed in your samples? Depending on pyroxenite composition, it may be possible to preserve cpx in residue at high degree of melting. Again this will help to test the robustness of your conclusion.
3. Nd-Hf isotope ratios: as you stated in L205-207, this may be beyond the scope of your study, but I think the isotope data can substantially strengthen your conclusion if indeed highly radiogenic Hf isotopic compositions are observed in your samples. Given the freshness of your samples, I suggest you do it (even not for this paper). The pay-off can be substantial. I will leave this an optional recommendation for you and the editor(s) to work out.

Additional comments and questions are listed below:

L45-46, delete 'anhydrous' and 'hydrous' here? Also, we don't know when plate tectonics starts
L60-64, this argument has been used in many studies, but a weak one. Because melt velocity in the melting column is considerably larger than the solid upwelling velocity, basalts erupted on the surface at MOR are always decoupled from their mantle source(s), as the melt and residue are segregated in the mantle and appear on the surface at different times.

L70: this should be a peritectic reaction, not eutectic.

L93, Table 2, show your data in a spider diagram? You could add a panel to Fig. S2. This will help your statement in L105-109.

L108, Fig. S4: colors are confusing here. I suggest you using either unfilled or light gray filled circle for Josephine samples. The green color makes it harder to see your new data. Also plot them in background, not foreground. (The same applies to other similar figures.)

L109-111, Fig. 3: some abyssal peridotite samples also have high Cr# (up to 60 in Fig. 3A), whereas Cr# of spinel in Josephine samples also extend to lower values (down to 15). What gives rise to the large spread? Also, not clear how useful Fig. 3B is. Plot Cr# vs. TiO₂ in spinel instead? Or Cr# in spinel vs. TiO₂ depletion in cpx?

Fig. 4. Caption (L350): blue squares – it's not blue color.

Table S2: where did you get the hydrous melting reaction? Bizimis et al. (2000)? Where did they get that melting reaction? Please provide a reference for the original study (i.e., that determined the melting reaction). Ref. 42 did not report a melting reaction.

L132-134, Fig. 4: solidus of hydrous peridotite is low, hence hydrous melting should start in the garnet stability field. It is essential to include this feature in your hydrous melting model here.

L139-142, Fig. S6: if you apply your melting models to opx, would the results (for T and Yb say) be consistent with those derived from your cpx modeling?

L156, 'artifact' a strange word here, do you mean anomaly?

L205, I hope you will measure Nd, Hf isotope ratios in at least some of your samples in a follow-up study.

L214: in your calculation, does subducted slab penetrate 660 km discontinuity? 60% seems a lot.

L216, where is Table 4?

L232, I disagree with this statement. Regardless where abyssal peridotites come from, they still represent mantle samples. Given the random and widely distributed nature of transform faults, the upper mantle beneath MOR is well sampled. It is the melting history for samples from transform fault and ridge axis that is different.

L241, delete 'far' or quantify your statement. Actually, water contents in fertile and depleted abyssal peridotites are fairly similar, at least they do not differ by orders of magnitude.

L246-262: this section reads like a summary. For implications, you could also move L239-245 to this section.

Supplementary tables: Most of them are unreadable to me (fonts too small). I would prefer you present them as Excel files for easy access.

Reviewer #2 (Remarks to the Author):

Line comments:

67: It would be helpful for the wider audience to briefly define fertile/refractory in the context of propensity for melting

148: Briefly describe what sweat veins are

157: What specifically about the Re-Os? TRD ages? Or whole rock isochrons?

180: explicitly make the connection for the reader about how positive shear wave anomalies are interpreted to be the result of dense material

182: define Vs

207: high Lu/Hf PAIRED with the inference from Re-Os that the melting event was ancient and NOT recent - be explicit

Figure 3: Why is the Josephine Peridotite included? It is mentioned in a supplementary figure, but not in the main body.

Figure 4: Yb - a heavy REE for the non-chemist readers

Figure S4: add a point/region with an estimate of DMM mantle (e.g. workman and hart) along with an arrow showing the direction of increasing melt depletion

We thank both reviewers for their thoughtful comments. We have addressed each below, line by line.

Reviewer #1 (Remarks to the Author):

Crust and mantle recycling are a natural consequence of plate tectonics and mantle convection, and an important topic in geochemistry, geodynamics, and seismology. Typical geochemical studies of recycling rely on isotope ratios and trace element abundances in basalts from ocean islands and mid-ocean ridges, and almost all these studies focus on the recycling of crust materials as they have larger isotopic (e.g., $^{87}/^{86}\text{Sr}$) and highly incompatible trace element (e.g., LILE) signals than mantle samples. Geochemical studies of the recycling mantle materials rely largely on Nd, Hf and Os isotope ratios and the main conclusion is that the recycled mantle materials are old and refractory.

There is no study that I am aware of that can pinpoint/deduce the tectonic setting of recycled mantle materials. The present study did just that by arguing that the ultra-depleted, clinopyroxene-bearing harzburgites at Mid Atlantic Ridge near 16°N representing a piece of recycled arc mantle. The authors' conclusion is based largely on similarities in major elements (Cr# in spinel, Mg# in olivine, Al_2O_3 in clinopyroxene) and selected incompatible trace elements (HREE and Ti) between their samples and peridotites from supra-subduction zone settings. They argue via trace element modeling that hydrous melting is required to explain the presence of residual clinopyroxene in the harzburgite and the very depleted Ti and HREE abundances in clinopyroxene (their Fig. 4). I generally agree with this argument and believe it is a novel study that will likely generate considerable interests in mantle geochemistry and geophysics communities. Nonetheless, I believe the modeling part can be improved (my main points#1 & #2 below) and additional data may help to strengthen or test the conclusion presented in this study (#2). Overall, the manuscript is well organized and easy to follow. I have a few suggestions on wording/structure that are listed in "Additional comments..." below. I would recommend this interesting manuscript for publication in NC after a round of revision.

Main Points:

1. Modeling of hydrous melting: The key figure used in study is Fig. 4 which compares Ti and Yb abundances in abyssal peridotites and SSZ peridotites with those predicted by anhydrous and hydrous melting. The modeling follows almost exactly the same method as Bizimis et al (2000), which is reasonable first step. But this is an old study. For example, it is not clear to me where Bizimis and coworkers got their hydrous melting reaction which you used in the present study. This is crucial in explaining clinopyroxene modal abundance and Ti-Yb depletion. Considerable progresses have been made in understanding hydrous melting since 2000 and such information should be included in the hydrous melting model in the present study. For example, the solidus of hydrous peridotite is considerably lower than the anhydrous one that hydrous melting will most

likely start in the garnet stability field. There are new melting reactions for hydrous melting, the most recent example is Grove and Till (CMP 2019). You could test a number of them to demonstrate the robustness of your conclusion.

We agree that garnet-facies melting is an important component to include, and have done so. We have also calculated Cpx-out degrees of melting for different melting modes (Table S3). We used numerous anhydrous melt modes from the literature including Baker et al. (1994), Kinzler (1997), and Wasylenki et al. 2003. Regardless of the anhydrous parameters used, we are unable to achieve the extents of melting required to satisfy trace element constraints from both Cpx and Opx before Cpx exhaustion. In addition, the starting trace element concentrations used (Workman and Hart 2005) are within uncertainties of other estimates, i.e. Salters and Stracke (2004), and thus do not change our conclusions, as the significant trace element depletions required are generated by the final few degrees of melting, as bulk D values decrease precipitously.

We thank the reviewer for pointing out the recent paper of Grove and Till (2019). We incorporated their results under garnet-facies conditions into our model, and tested both their High T (1075-1125 °C) and Low T (950°C-1050°C) melt modes. Using Ti and HREE concentrations in Cpx, we were able to use a residual sum of the squares minimization to solve for extents of melting in both the garnet and spinel stability field, correcting phase proportions across the garnet-spinel transition after the method of Johnson et al. (1990). Our results require ~6-8% melting in the garnet field, followed by >20% melting in the spinel field, degrees of melting which have not been observed at mid-ocean ridge settings. Interestingly, our calculated modes agree quite well with sample modes determined by point counting, indicating that 16°N peridotites were very near the point of Cpx exhaustion (Table S3). This model has been included in our manuscript for examination by the reviewers.

2. Mantle source composition: just curious, if you melt a garnet pyroxenite or a pyroxenite-peridotite mixed source, would it be possible to produce the ultra-depleted cpx observed in your samples? Depending on pyroxenite composition, it may be possible to preserve cpx in residue at high degree of melting. Again this will help to test the robustness of your conclusion.

This is a good point. Pyroxenite veins are commonly found in ophiolites, however very few, if any, true pyroxenites (not derived from a cumulate origin) have been recovered from mid-ocean ridges. Cumulate pyroxenites have been noted along the SWIR, attributed to deep crystallization (>0.6 GPa) of N-MORB like melts (Dantas et al. 2007). This is likely due to the lower solidus temperature of pyroxenites, which are thought to melt entirely prior to peridotite emplacement on the seafloor. Recent modeling by Brunelli et al. (2018) has shown at the VEMA lithospheric section that the presence of pyroxenites in a plum pudding mantle actually inhibits melting of the surrounding peridotite, due to the transfer of the latent heat of fusion required to melt the low-solidus pyroxenite from adjacent peridotite. We did model the melting of a garnet-clinopyroxene assemblage using a 25-75% starting mode, with partition coefficients, melting modes, and initial concentrations of Pertermann et al. (2004). Only at melting in excess of 90% are similar Cpx Ti concentrations observed, however at such extreme extents of melting MREE and HREE concentrations approach zero (sub ng/g), contrary to observed concentrations. Thus we find a pyroxenite protolith unlikely. In addition, textural

evidence (e.g. Fig. S1) precludes a pyroxenite origin for 16°N Cpx.

3. Nd-Hf isotope ratios: as you stated in L205-207, this may be beyond the scope of your study, but I think the isotope data can substantially strengthen your conclusion if indeed highly radiogenic Hf isotopic compositions are observed in your samples. Given the freshness of your samples, I suggest you do it (even not for this paper). The pay-off can be substantial. I will leave this an optional recommendation for you and the editor(s) to work out.

We agree that Hf isotopes would be useful. We have consulted with two isotope geochemists (V. Salters, FSU and C. Liu, Institute for Geology and Geophysics, CAS) to discuss this. Hf concentrations in these Cpx are >4 ng/g on average, less than that required for isotope analyses (Table 2). We hope analytical advances will allow this work in the future.

Additional comments and questions are listed below:

L45-46, delete 'anhydrous' and 'hydrous' here? Also, we don't know when plate tectonics starts

We have deleted 'anhydrous' and 'hydrous' from this sentence, and have simplified the text (removed 4.5 Ga) to simply state "over Earth's history". We do not wish to provoke the "when did modern plate tectonics begin" debate here!

L60-64, this argument has been used in many studies, but a weak one. Because melt velocity in the melting column is considerably larger than the solid upwelling velocity, basalts erupted on the surface at MOR are always decoupled from their mantle source(s), as the melt and residue are segregated in the mantle and appear on the surface at different times.

We agree that melt velocity is typically orders of magnitude larger than peridotite upwelling velocity, and thus basalt and peridotite compositions may not correlate. We have amended our text to reflect this point (L. 78).

L70: this should be a peritectic reaction, not eutectic.

We have corrected this, thank you for catching the error.

L93, Table 2, show your data in a spider diagram? You could add a panel to Fig. S2. This will help your statement in L105-109.

We have added a full spider diagram of 16°N Cpx, which we have added to Fig. S5 (formerly Fig. S6).

L108, Fig. S4: colors are confusing here. I suggest you use either unfilled or light gray filled circle for Josephine samples. The green color makes it harder to see your new data. Also plot them in background, not foreground. (The same applies to other similar figures.)

We agree that the colors were confusing. We have reconstructed Fig. S4, using white squares for the Josephine and bringing our 16°N data to the foreground.

L109-111, Fig. 3: some abyssal peridotite samples also have high Cr# (up to 60 in Fig.

3A), whereas Cr# of spinel in Josephine samples also extend to lower values (down to 15). What gives rise to the large spread? Also, not clear how useful Fig. 3B is. Plot Cr# vs. TiO₂ in spinel instead? Or Cr# in spinel vs. TiO₂ depletion in cpx?

This is an astute observation. The spread in Cr# for Josephine covaries with other proxies for degrees of melting, e.g. Cpx modal abundances and Al₂O₃ in Cpx and Opx. This variability likely reflects the availability of fluids during hydrous flux melting within the mantle wedge, where localized delivery of subduction-derived fluids may promote highly heterogeneous melting extents over length scales of kilometers. A more detailed discussion may be found in Le Roux et al. (2014) and references therein.

Fig. 4. Caption (L350): blue squares – it's not blue color.

We have corrected the name to cyan squares.

Table S2: where did you get the hydrous melting reaction? Bizimis et al. (2000)? Where did they get that melting reaction? Please provide a reference for the original study (i.e., that determined the melting reaction). Ref. 42 did not report a melting reaction.

We used the preferred melting reaction of Bizimis et al. (2000), which originated from Gaetani and Grove (1998). We have elaborated on modeling details in Table S3 so that our results may be replicated in their entirety.

L132-134, Fig. 4: solidus of hydrous peridotite is low, hence hydrous melting should start in the garnet stability field. It is essential to include this feature in your hydrous melting model here.

We agree with the reviewer, and have added garnet-facies melting to our models to more thoroughly establish and justify our conclusions.

L139-142, Fig. S6: if you apply your melting models to opx, would the results (for T and Yb say) be consistent with those derived from your cpx modeling?

We conducted additional Opx TE modeling to answer this. Using an identical model to our Cpx calculations, for D41-24 our Opx Ti and Yb modelled concentrations (35.1 and 0.057 µg/g, respectively) broadly agree with observed concentrations (41 and 0.0958 µg/g, respectively). Modelled Opx TE abundances were always lower than observed, likely due to uncertainties in partition coefficients, melting conditions, and perhaps melt-rock reaction. Thus, our modeling results are consistent with both Cpx and Opx TE abundances.

L156, 'artifact' a strange word here, do you mean anomaly?

We agree that the term artifact is odd, and have replaced it with relict, a word that more accurately conveys our intended meaning.

L205, I hope you will measure Nd, Hf isotope ratios in at least some of your samples in a follow-up study.

We plan to do so, and have reached out to potential collaborators in this endeavor.

L214: in your calculation, does subducted slab penetrate 660 km discontinuity? 60% seems a lot.

We only consider the volume of mantle above the 660 km discontinuity for our calculations, and below 30 km from the surface. We agree that this value is high, yet it is similar to those observed by Simon et al. (2008) and calculated by Salters et al. (2011). Our mantle processing rate ($77 \text{ km}^3/\text{yr}$) may be overestimated; as we note in the text, both adiabatic and hydrous flux melting likely occur at sub-arc depths, whereas only the latter would produce such highly depleted peridotites.

L216, where is Table 4?

We thank the reviewer for catching this numbering error. We were referring to Table 3, and have corrected the manuscript.

L232, I disagree with this statement. Regardless where abyssal peridotites come from, they still represent mantle samples. Given the random and widely distributed nature of transform faults, the upper mantle beneath MOR is well sampled. It is the melting history for samples from transform fault and ridge axis that is different.

We have removed this statement from the text. We note that significant portions of the MOR system have yet to be systematically dredged outside the immediate axial volcanic zone, as many cruises sought only fresh basalt glasses. Thus, additional coverage would be useful to examine spatial distributions of variably depleted peridotites along MORs.

L241, delete 'far' or quantify your statement. Actually, water contents in fertile and depleted abyssal peridotites are fairly similar, at least they do not differ by orders of magnitude.

We have deleted the word far; as the reviewer accurately points out, water abundances are quite similar in mantle rocks analyzed to date. This is at odds with experimental studies which have shown that water is highly incompatible in mantle assemblages (e.g. Hauri et al. 2006). This contradiction may be the result of the rapid reequilibration of hydrogen during peridotite emplacement. Regardless, it does require further investigation.

L246-262: this section reads like a summary. For implications, you could also move L239-245 to this section.

We have moved L239-245 to the implications, and removed the first few lines of summary text from the Implications section.

Supplementary tables: Most of them are unreadable to me (fonts too small). I would prefer you present them as Excel files for easy access.

We agree, and apologize for this inconvenience. We were unable to upload our tables in Excel format during submission, but would be happy to send the files as Excel sheets now or in a second review.

Reviewer #2 (Remarks to the Author):

Line comments:

67: It would be helpful for the wider audience to briefly define fertile/refractory in the context of propensity for melting

We have added a line to this effect in the introduction (L.38).

148: Briefly describe what sweat veins are

We have removed the word sweat to avoid confusion. In essence, these orthopyroxenites are the final late liquids produced during sub-arc melting, which crystallize to form veins.

157: What specifically about the Re-Os? TRD ages? Or whole rock isochrons?

Here we refer to single sulfide grain ages (2.06 ± 0.26 Ga) of Harvey et al. (2006). We revised the text to reflect these ancient ages (L. 49).

180: explicitly make the connection for the reader about how positive shear wave anomalies are interpreted to be the result of dense material

We have explicitly stated the connection for readers (L. 278).

182: define Vs

We now define Vs in the manuscript (L. 266-269).

207: high Lu/Hf PAIRED with the inference from Re-Os that the melting event was ancient and NOT recent - be explicit

We have added this coupled requirement of *both* high Lu/Hf and ancient melting event ages (L. 331-334). We thank the reviewer for pointing this out.

Figure 3: Why is the Josephine Peridotite included? It is mentioned in a supplementary figure, but not in the main body.

We now explicitly mention the Josephine Peridotite in the text (L. 148).

Figure 4: Yb - a heavy REE for the non-chemist readers

We have added "a heavy REE" to the caption of Figure 4.

Figure S4: add a point/region with an estimate of DMM mantle (e.g. workman and hart) along with an arrow showing the direction of increasing melt depletion

We have added a white star denoting DMM mantle (bulk) as well as a melt depletion trendline.

REVIEWERS' COMMENTS:

Reviewer #1 (Remarks to the Author):

This revised MS addressed all the major concerns raised in my earlier review. I am happy to recommend it for publication in its present form, although I would encourage the authors to redrawn Fig. 4: here the blue squares and circles produced by your hydrous melting model are too many and confusing. I would change them into tick marks. Also the dashed line will not be visible after 50% reduction of this figure. A very nice paper.

Reviewer Comments and Author Response

Reviewer #1: This revised MS addressed all the major concerns raised in my earlier review. I am happy to recommend it for publication in its present form, although I would encourage the authors to redraw Fig. 4: here the blue squares and circles produced by your hydrous melting model are too many and confusing. I would change them into tick marks. Also the dashed line will not be visible after 50% reduction of this figure. A very nice paper.

BU: We have redrafted Figure 4 to enhance readability, using ticked lines instead of circles. We have also made all SSZ peridotite data the same format, to simplify the figure.

Yours sincerely,

Benjamin M. Urann and co-authors